# Elicitors Modulate Young Sandalwood (*Santalum album* L.) Growth, Heartwood Formation, and Concrete Oil Synthesis

**DOI:** 10.3390/plants10020339

**Published:** 2021-02-10

**Authors:** Yuan Li, Xinhua Zhang, Qingwei Cheng, Jaime A. Teixeira da Silva, Lin Fang, Guohua Ma

**Affiliations:** 1Key Laboratory of South China Agricultural Plant Molecular Analysis and Genetic Improvement, South China Botanical Garden, Chinese Academy of Sciences, Guangzhou 510650, China; liy@scib.ac.cn (Y.L.); xhzhang@scib.ac.cn (X.Z.); chengqw0820@foxmail.com (Q.C.); linfang@scbg.ac.cn (L.F.); 2University of the Chinese Academy of Sciences, Beijing 100049, China; 3Independent Researcher, P.O. Box 7, Miki-cho Post Office, Ikenobe 3011-2, Kagawa-Ken, Miki-cho 761-0799, Japan; jaimetex@yahoo.com

**Keywords:** concrete oil, exogenous elicitors, heartwood formation, sandalwood

## Abstract

Five chemical elicitors––6-benzyladenine (BA), ethephon (ETH), methyl jasmonate (MeJA), hydrogen peroxide (H_2_O_2_) and calcium chloride (CaCl_2_)––were used to treat 1- and 5-year-old sandal trees (*Santalum album* L.) to assess their effects on growth, heartwood formation and concrete oil synthesis. The results showed that some newly formed branches in stems that were induced by BA and ETH displayed leaf senescence and developed new smaller and light-green leaves. The relative percentage of concrete oil from the heartwood of water-treated trees (0.65%) was significantly lower than that from trees treated with 4 mM H_2_O_2_ (2.85%) and 4 mM BA (2.75%) within one year. Four mM BA, H_2_O_2_ and CaCl_2_ induced a significantly higher level of sesquiterpenoids than heartwood treated with 2 mM of these elicitors. Four mM MeJA induced significantly less sesquiterpenoids than heartwood treated with 2 mM MeJA. Morphological, physiological, and chromatographic–spectrometric technologies were integrated to trace the potential function of these exogenously applied chemical elicitors. The results may have important applications and provide a better understanding of the molecular mechanism of heartwood formation and hardening in young sandalwood trees.

## 1. Introduction

*Santalum album* L. (Indian sandalwood) has great commercial value for its fragrant essential oil and its valuable carving wood [1,2]. However, sandal trees cannot be compared to other commercial short-rotation or timber-yielding species because slow-growth and over-exploitation hamper productivity [3]. In India, sandal trees begin to form heartwood at the age of 10 to 13 years with the fastest growth period at 20 years in natural stands [4]. Moreover, global sandalwood resources have diminished since 1980, and demand is high [5]. In the long run, expanding plantations could alleviate the need to harvest natural populations but stringent conservation efforts are needed to thwart the decline of natural resources [6]. Enhancement of heartwood formation, especially through increasing plantation productivity, is an important means for creating sustainable sandalwood resources.

Many tree species form central heartwood in the inner core layers of the wood, which has the ability to accumulate secondary metabolites such as phenolics and terpenes. Heartwood was commonly considered to be dead tissue in established heartwood formation of model plants *Robinia pseudoacacia* and *Juglans* species [7,8,9,10]. Sandalwood trees have been proposed as an additional model of heartwood formation, which implies that in heartwood at least a small number of living parenchyma cells become specialized for terpene biosynthesis, which was validated by sandalwood functional genomics and metabolite analyses [11,12]. Various heartwood terpenes have been explored for the fragrance industry. Sandal essential oil contains various sesquiterpenols such as *α*- and *β*-santalol, *epi*-*β*-santalol and *α*-exo-bergamotol [11]. Of these, *α*- and *β*-santalol contribute over 80% of the total essential oil. Therefore, effective ways to improve the production of high-value bioproducts in heartwood or the yield of essential oils in sandalwood production are needed. Many chemical elicitors are critical for plant growth and development and play an important role in controlling various signals and downstream responses by modulating various transporters and biochemical reactions [13]. Several studies reported that terpenes were induced by various elicitors and signal stimuli including calcium ion, abscisic acid, polyamines, methyl jasmonate (MeJA), nitric oxide, herbicides, and wounding [13,14]. In the *Santalum* species, a few field experiments focused on inducing heartwood formation. Those studies found that the mechanical damaging of tree stems alone did not have obvious effects on heartwood formation while the yield of heartwood or essential oil could be induced by mechanical damage and chemical elicitors. The exogenous injection of a solution of CuSO_4_ and ZnSO_4_, combined with indole-3-butyric acid (IBA), into the stems of 12-year-old sandal trees formed heartwood [15]. However, that study lacked a qualitative and quantitative analysis of essential oil content and composition. Some studies focused on different concentrations of 6-benzyladenine (BA) and ethephon (ETH) by injecting or girdling stems of trees of different ages [16,17]. Spraying leaves of 10-month-old sandal trees with BA at 1.0 mg/L promoted stem growth, crown width and dry biomass accumulation; enhanced the net photosynthetic rate and instantaneous water use efficiency; and stimulated photosynthetic pigment. However, height was considerably inhibited [18]. ETH increased heartwood formation but it inhibited vegetative growth [16,19]. The herbicide paraquat was able to extend the heartwood area [19]. Some studies described the effects of a few chemical elicitors on physiological formation of heartwood in sandalwood [16]. Little information on the potential function and mechanism of exogenously applied chemical elicitors in relation to plant growth, heartwood formation, and concrete oil accumulation is available when physiological and anatomical analyses are combined. In this study, we investigated the effect of five chemical elicitors––BA, ETH, MeJA, hydrogen peroxide (H_2_O_2_) and calcium chloride (CaCl_2_)––on growth, heartwood formation, and concrete oil synthesis of young (1- and 5-year-old) sandalwood trees by integrating anatomical, physiological, and mass-spectrum evidence.

## 2. Materials and Methods

### 2.1. Plant Growth Conditions and Selection of Materials

*Santalum album* seeds were first soaked in 2 mM gibberellic acid for 12 h. Then the seeds were sown in sand in a nursery basin at 27–32 °C for at least 4 weeks. After the seeds germinated and the seedlings developed at least four leaves, 10 cm tall seedlings were transplanted into 14 × 11 cm plastic pots filled with a mixture of loess and humus (5:1, *v/v*), and placed in a shaded shed. After 1 month, the seedlings parasitized a host species *Kuhnia rosmarinifolia* Vent. As the seedlings grew to over 30 cm in height, the seedlings and their host species were transferred to 25 cm × 30 cm cloth bags filled with a mixture of loess and humus (5:1, *v/v*) [20]. After cultured for 1 year, 40 healthy sandal trees with consistent growth were randomly selected for the field experiment. All the sandal trees were grown in a glasshouse in South China Botanical Garden, Chinese Academy of Sciences (23°10′ N, 113°21′ E).

For the field experiment, the 5-year-old sandal trees parasitized another host species, *Tephrosia candida* (Roxb.) DC. The field experiment was performed in Huangjiang town, in Dongguan city, Guangdong province, China (22°54′ N, 114°01′ E). Trees were all planted in 2012 and separated by a distance of 3 m × 4 m. Five-year-old sandal trees, all with a 6–7 cm diameter at breast height (DBH) (130 cm above ground) were selected. From among them, trees without heartwood were further selected according to the methods of Radomiljac [19] with a minor modification. Stems at 30 cm above the ground were sampled by drilling a hole that sloped down at a 45° angle to the trunk, using a 12 V electric drill with a 5 mm drill diameter (Oulipu, Shanghai, China) to observe the color and fragrance of chips. A total of 44 healthy 5-year-old trees, characterized by consistent growth, 6–7 cm DBH, and without heartwood were selected for the field experiments. To avoid additional damage to the trees by drilling, in the next experiment, chemical elicitors were injected into the same 5 mm holes that were drilled into the trunk.

### 2.2. Treatment with Exogenous Chemical Elicitors

Five chemical elicitors were selected for this study: BA (>99%) (EKEAR Bio, Shanghai, China), ETH (>85%) (Solarbio, Beijing, China), MeJA (>95%) (Aladdin, Shanghai, China), H_2_O_2_ (analytic reagent, AR) (Sinopharm Chemical Reagent Co., Shanghai, China), and CaCl_2_ (>99%) (Chembase, Beijing, China). The chemicals (ETH, H_2_O_2_, and CaCl_2_) were dissolved in water. BA was dissolved in sodium hydroxide (NaOH) (>97%) (Aladdin), then diluted with water to achieve the required volume. MeJA was dissolved in Tween-20 (Macklin, Shanghai, China), then diluted with water to achieve the required volume. Since the stems of 1-year-old sandalwood trees are thin, these chemical elicitors were added to trees by spraying leaves [21] with a few modifications. A single spray of 100 mL of 2 mM BA, ETH, MeJA, H_2_O_2_ or CaCl_2_ was applied to the abaxial and adaxial surfaces of leaves of 1-year-old sandalwood trees. An equal volume of water, NaOH, and Tween-20 aqueous solutions were added to 1-year-old trees serving as the controls. To eliminate, as much as possible, the volatility of sprayed chemical substances on other trees in different treatments, especially MeJA and ETH, which volatilize easily, five seedlings of each treatment were moved to adjacent glasshouses where 2 mM MeJA or ETH was exogenously applied to their leaves. Since the other chemical elicitors have lower volatility, trees were spaced 5 m apart in the same glasshouse. One day after the application of MeJA and ETH, treated trees were moved to the glasshouse with trees treated with other chemical elicitors, and also spaced 3 m apart [21]. Air flowed upward in the glasshouse. There were no differences in plant height, stem xylem width, mature stem diameter and stem tissue structure among trees treated with water, NaOH, or Tween-20 aqueous solutions. Consequently, water was selected as the control for the next experiment.

For the field experiment, 2 mM and 4 mM of these chemical elicitors, with water as the control, were injected as a liquid solution into the 5 mm diameter of holes drilled into the stems of 5-year-old trees at 30 cm above the ground. In each injection, 400 mL of each chemical (or water) was stored in an infusion bag and transfused four times a year by injection into drilled holes. To avoid additional damage, desiccation or contamination, plastic stoppers were added to the drilled holes after the injection of each solution.

### 2.3. Anatomical Observations

For anatomical observations, approximately 1 cm-long stems at the 20th internode from the shoot tip (about 30 cm above the ground) of 1-year old trees treated with 2 mM of any of the five chemical elicitors as well as the control were fixed in FAA, that is, a mixture of formalin (AR 37%), acetic acid (AR > 99%) and alcohol (ethanol, AR > 99%) (70% alcohol–formalin–acetic acid = 90:5:5, *v/v*) (Shanghai Macklin Biochemical Co. Ltd., Shanghai, China). The stems were sectioned by hand to a thickness of about 100 μm. After washing, dehydrating, and clearing, the sections were added to a solution of pure methyl salicylate (AR > 99%) (Shanghai Macklin Biochemical Co. Ltd., Shanghai, China), then observed under a Leica DVM6 3D digital microscope (Leica microsystem, Wetzlar, Germany).

### 2.4. Plant Height, Stem Diameter, Xylem Width, DBH, and Heartwood Extension Measurements

In both 1- and 5-year-old trees, the height of trees treated with chemical elicitors or the control was recorded using a 7 m telemeter rod at maximum range. The diameter of the 20th internode from the shoot tip (about 30 cm above the ground) was recorded using a vernier calliper. Xylem width in 1-year-old trees was measured from sections using Digimizer software (version 5.4) with five sections per sample. The diameter of stems at a distance of 30 cm above the ground (position of injection of chemical elicitors or water) in 5-year-old trees and DBH at a distance of 130 cm above the ground in 5-year-old trees was measured (circumference/π) with a soft tape measure after the exogenous application of chemical elicitors or water. For heartwood, the longitudinal extended distance (LED) was measured with a minor modification. To achieve this, an electric drill was used to drill sandalwood chips at a distance of every 5 cm from the fluid infusion point to distinguish the change in color from white to brown, and to assess the emission of a fragrant smell. These experiments were carried out in a randomized block design with 5 or 4 independent biological replicates in 1- and 5-year-old trees, respectively.

### 2.5. Concrete Oil Extraction from Heartwood

Heartwood samples were collected from treatment and control trees with a tree growth cone (Haglof, Långsele, Sweden). Air-dried heartwood was weighed and used to extract concrete oil. At first, the heartwood was cut into small pieces then ground into a powder with an electric grinder. Concrete oil was extracted using 0.5 g of heartwood powder at a 1:15 (*w/v*) ratio with anhydrous diethyl ether (Baishi Chemical Industry Co., Tianjin, China) in a 100 mL glass bottle with minor modifications [22]. After ultrasonication (200 W, 40 Hz) for 45 min in an ultrasonicator (SD5200DTN, Ningbo, China), the extract was left standing for 3 days, and then dehydrated with anhydrous sodium sulfate (AR > 99%) (Aladdin) overnight. After the first filtrate was filtered through a 0.22 μm filter membrane (Jinteng, Tianjin, China), the residue of extracted concrete oil was added at a ratio of 1:10 (*w/v*) to anhydrous diethyl ether to extract for an additional 3 days. After the second filtration, the first and second filtrates were pooled and preserved in a microcentrifuge tube. Diethyl ether was evaporated to dryness by using a Termovap sample concentrator (ECOM, Trebonicka, Czech Republic) and weighed. The total extracted concrete oil content was calculated as the relative percentage of oil weight to dried heartwood weight (*w/w*).

### 2.6. GC-MS Analysis

The extracted concrete oil was diluted to 1.0% (or 1:100) (*v/v*) with anhydrous diethyl ether prior to analysis. GC-MS was performed on a GCMS-QP2010PLUS (Shimadzu, Hadano, Japan) equipped with a HP-5MS column (Agilent Technologies, Santa Clara, CA, USA) (30 m long, 250 μm internal diameter, 0.25 μm film thickness). The oven program comprised initial 50 °C, a ramp of 5 °C min^−1^ to 260 °C, 260 °C for 2 min, a ramp of 20 °C min^−1^ to 300 °C, then 300 °C for 15 min. The injector temperature was maintained at 250 °C. The carrier gas was helium at a flow rate of 1 mL min^−1^ and 1.5 μL injections (split 1:50) made by an autosampler. Constituents were detected by mass spectra fitted with an EI source operated at 70 eV with a source temperature of 250 °C. Mass spectra were recorded in the range 20–550 *m/z* at 2 scans s^−1^ and dwell time was 2.5 min. Using these conditions, a total ions chromatogram of sandalwood concrete oil and mass spectrometric data were obtained. LabSolutions GC-MS solution software (Shimadzu) was used for data acquisition and processing. Compounds were identified by comparison of mass spectra against four National Institute of Standards and Technology (NIST, Gaithersburg, MD, USA) standards: NIST2005, NIST2005s, NIST2014, and NIST2014s. In addition, compounds were also identified using authentic sandalwood essential oil standards (batch number 110789-200005; National Institutes for Food and Drug Control, Beijing, China) by comparing their corresponding chromatographic peak retention indices under the same injection and analysis conditions. On the basis of the total ions chromatogram, relative quantitative analysis of the extracted concrete oil components and internal standard dodecane was performed using normalized peak areas and dried heartwood weight.

### 2.7. Statistical Analysis and Sample Replicates

Data of plant height, stem diameter, xylem width, DBH, heartwood extension measurements, and concrete oil volume and constituents were analyzed in SPSS v. 20.0 (SPSS Inc., Chicago, IL, USA). Following analysis of variance to separate treatment means, a student’s *t*-test or Duncan’s multiple range test was used to assess significant differences between treatment pairs or multiple treatments, respectively (α = 0.05).

## 3. Results

### 3.1. Effects of the Elicitors on Stem Cambium Cell Activity in 1-Year-Old Trees

There were significant differences in plant height, xylem width, and mature stem diameter in trees treated with all five chemical elicitors compared with water-treated trees (Figure 1a–c). The trees treated with H_2_O_2_, BA, and ETH were approximately 26.1%, 16.4% and 8.91% taller than those treated with water while the trees treated with ETH, MeJA, and CaCl_2_ showed no significant differences relative to the control (Figure 1a). Compared with water-treated trees, the stem width was obviously enlarged, nearly 1.5 times wider, especially in BA- and H_2_O_2_-treated trees (Figure 1b). Next, a series of stem sections was made to observe anatomical structure. Consistent with stem diameter data, the regions of cambium-derived xylem were also significantly wider and an increase in cell layers of xylem tissue mainly contributed to the enhanced stem width (Figure 1c,d). As seen from results related to stem sections (Figure 1d), the ability of secondary xylem to differentiate was induced by all five chemical elicitors, suggesting that they could accelerate the process of wood formation in young trees. This data, when combined with the results of plant height, indicates that H_2_O_2_ and BA are the two most effective chemical elicitors among the five tested, to induce the growth of 1-year-old sandal trees.

### 3.2. Effects of the Elicitors on Morphology, Height and Stem Diameter Changes of 5-Year-Old Sandalwood Trees

Some newly formed branches in stems were induced by BA (Figure 2a) and the treatment with ETH resulted in leaf senescence. Those branches then developed smaller and fewer new light-green leaves (Figure 2b). However, other trees treated with MeJA, H_2_O_2_ and CaCl_2_ did not show any obvious side-effects or positive effect on tree morphology after one year of observation.

As shown in Table 1, compared with other treatments, 4 mM ETH resulted in the least change in diameter at 30 and 130 cm above the ground, while 4 mM CaCl_2_ resulted in the highest diameter at 130 cm above the ground (Table 1). There were no significant differences in stem diameter between different concentrations of BA, MeJA, H_2_O_2_, and water (Table 1).

A significant change was found between the height of trees treated with 4 mM BA, 2 mM ETH and 2 mM MeJA compared to trees before treatment (Figure 3a). Compared with the water control, no significant differences in plant height were observed among the five chemical elicitors (Figure 3b).

### 3.3. Chemical Elicitors Induced Heartwood Formation

Within 2 months, a pale-yellow color and fragrant smell appeared in the heartwood of trees treated with 4 mM BA. However, the control trees had not yet formed any heartwood (Figure 4a). BA induced significantly higher heartwood LED (average increase: 135–141 cm) than other treatments (average increase: 6.25–25 cm) (Figure 4c). Heartwood in trees of all treatments was successfully induced within 1 year. Heartwood in H_2_O_2_-treated trees turned an obvious pale brown and was concentrated while that of other treatments appeared dark yellow and was scattered over several points, similar to the water control (Figure 4b).

### 3.4. H_2_O_2_, BA and MeJA Affect Concrete Oil Content and Its Constituents

After 1 year, the concrete oil from all treatments was extracted and collected in tubes (Figure 5). The concrete oil (0.65%) extracted from the heartwood of water-treated trees was significantly less than the 2.85%, 2.75%, 1.76%, and 1.58% from trees treated with 4 mM H_2_O_2_, 4 mM BA, 2 mM BA, and 2 mM MeJA, respectively (Figure 6). In contrast, CaCl_2_, ETH, 2 mM H_2_O_2_, and 4 mM MeJA did not increase concrete oil content compared with the water control (Figure 6).

The major sesquiterpenoid constituents in sandal concrete oil were identified through GC-MS (Figure 7). We focused on the major sesquiterpenoid constituents (Figure 8). *α*-, *β*- and *epi*-*β*-santalol, *trans*-*α*-bergamotol, and *α*-, *β*- and *epi*-*β*-santalene were all detected in heartwood treated by the five chemical elicitors or water, whereas 2 mM BA, 4 mM BA, 2 mM MeJA, and 4 mM H_2_O_2_ significantly increased sesquiterpenoid content, 2 mM CaCl_2_ significantly decreased the content of these seven compounds compared with the water control (Figure 8a–g). *Trans*-*α*-bergamotene was not detected in the heartwood of trees treated with water, ETH, CaCl_2_ or 2 mM H_2_O_2_, but it was induced after the application of BA, MeJA and 4 mM H_2_O_2_ (Figure 8h). ETH did not increase these sesquiterpenoids compared with the water control (Figure 8a–h).

The concentrations of the five chemical elicitors had an obvious influence on sandal concrete oil synthesis. Changes to concrete oil synthesis were divided into different outcomes (Figure 8): 4 mM BA, H_2_O_2_, and CaCl_2_ induced a significantly more sesquiterpenoids than 2 mM BA, H_2_O_2_, and CaCl_2_ while 4 mM MeJA induced significantly fewer sesquiterpenoids than did 2 mM MeJA. There were no significant differences in the quality of the concrete oil from heartwood treated with either 2 or 4 mM ETH.

## 4. Discussion

### 4.1. Stem Cambium Cell Activity and Heartwood Formation: Response to Chemical Elicitors

Anatomical evidence showed significant differences in stem diameter and width of the xylem region between treatments and the water control in 1-year-old trees, suggesting that vascular cambium activity was quickly induced by the five chemical elicitors at different concentrations, especially BA and H_2_O_2_ (Figure 1). These findings support the action of cytokinins, which can regulate vascular cambium activity [23]. A reduced level of cytokinin by overexpressing the cytokinin-degrading enzyme cytokinin oxidase (*CKX*) led to a decrease in the number of cambium cells and reduced stem diameter in poplar [24]. There may be two functions of H_2_O_2_ in regulating cambium activity: the first by acting as a signaling molecule and the other by functioning in crosstalk with phytohormone signaling when applied at a low concentration [25]. For example, the first transcriptome-wide analyses of plant H_2_O_2_ signaling indicated a close cross-talk with ethylene [26]. During shoot development, H_2_O_2_ can mediate apical dominance and axillary bud formation by affecting auxin and cytokinin homeostasis [27]. Another explanation is that H_2_O_2_ plays a role in regulating cell-wall loosening by serving as a substrate for peroxidases, as an important precursor of the hydroxyl radical, by cleaving cell-wall xyloglucan polymers and prematurely promoting secondary cell wall formation [28,29]. Since wood differentiates from the vascular cambium, we believe that exogenously applied BA and H_2_O_2_ contributed to an increase in the activity of the vascular cambium, and that this might determine the process of wood formation. These assumptions were partly verified by the changes to heartwood in 5-year-old trees after treatment with chemical elicitors.

In some tree species, the generation of yellow or brown extracts and the appearance of a fragrant scent is a simpler and more convenient way to determine if heartwood has formed rather than measuring heartwood formation using programmed cell death or specific enzyme activity [30,31,32,33]. In BA-treated trees, the change in color and emission of a scent in wood chips demonstrated that BA is an effective chemical elicitor for inducing heartwood in 5-year-old sandalwood trees (Figure 4a,b). Furthermore, an average of 140 cm of heartwood LED demonstrates the high efficiency of BA to form heartwood. From the perspective of a heartwood harvest, exogenously applied BA is a beneficial method to promote heartwood prematurely and enhance its production in young sandalwood. In addition to BA, heartwood LED in trees treated with other chemical elicitors extended no more than 30 cm above the injection point (Figure 4c). This could be explained by vast methodological differences, including the type, concentration, and volume of elicitor applied; treatment period; and tree age.

### 4.2. Response of Plant Height and Diameter to Chemical Elicitors

In this study, the degree of change to tree height and trunk diameter differed in 1-year-old (early vegetative period) and 5-year-old (late vegetative period) sandal trees after treatment with the same chemical elicitor (Figure 1a,b and Figure 3, Table 1). One plausible explanation for the larger growth rate (the maximum was nearly 26% higher and 1.5-fold wider than the water control, Figure 1a,b) in 1-year-old than in 5-year-old trees (no significant change compared with water control, Figure 3b, Table 1) might be that 1-year-old trees are in a period of rapid growth while 5-year-old trees may have entered a slower growth stage. The effect of chemical elicitors on the height and diameter of older 5-year-old sandalwood trees was not clearly seen in our study, so additional studies that are able to trace changes at the cellular level are needed to make a more comprehensive interpretation.

### 4.3. Essential Oil Synthesis in Response to BA and H_2_O_2_

In Guangdong province, although approximately 15% to 20% (3/20, 4/20, and 4/20) of natural stands of 6-year-old sandalwood trees could form heartwood, their relative concrete oil content was less than 1.02% (0%, 0.82%, and 1.02%). Our average 2.85%, 2.75%, and 1.76% concrete oil yield (Figure 5) for 4 mM H_2_O_2_, 4 mM BA, and 2 mM BA, respectively, shows that these elicitors could effectively induce concrete oil in 5-year-old sandalwood trees at levels similar to 10-year-old sandalwood trees [16]. Moreover, the main identified sesquiterpenes confirmed that both H_2_O_2_ and BA play important roles in the synthesis of sesquiterpenes of sandal concrete oil.

Misra and Dey [34] found that *α*-santalol was localized in secondary xylem, parenchymatous ray cells, cortical parenchyma and the epidermis of mature stems of 15-year-old sandalwood trees. The enhanced content of santalols may be related to the rapid formation of heartwood induced by chemical elicitors, especially BA and H_2_O_2_. Similarly, the application of BA-enhanced volatile oil content in *Lavandula dentata* [35]. Thus, BA plays a role in heartwood formation and synthesis of sesquiterpenes, consistent with our results for concrete oil yield and content of sesquiterpenes, indicating that sandal oil synthesis may be related to BA (Figure 6 and Figure 8). Exogenously applied H_2_O_2_ enhanced the induction of sesquiterpene synthase and triggered the production of jasmonate and salicylic acid-induced agarwood sesquiterpene accumulation in the stems of *Aquilaria sinesis* [36]. The function of BA in sandal oil synthesis depends on crosstalk with other phytohormones or transcription factors that regulate key downstream genes coding for sesquiterpene synthase, resulting in the accumulation of sesquiterpenes although this needs to be verified in a more detailed investigation using molecular and genetic methods.

## 5. Conclusions

The main commercial value of mature sandalwood lies in its aromatic heartwood and high level of santalol extracted from its heartwood. The ultimate goal of planting sandalwood is to harvest high-quality heartwood. This study developed a valid way of improving heartwood value as early as possible without devastating or damaging natural stands of sandal trees. In our study, by drilling a small hole in young sapwood first emitted a wounding signal in trees. However, the slight changes caused by the injection of water to drilled holes on heartwood formation confirmed that mechanical damage did not obviously impact the induction of heartwood. Therefore, compared with wounding injection with water, the shortened cycle of heartwood formation, increased level of volatile essential oil, and higher production of main sesquiterpenes suggests that the use of H_2_O_2_ and BA as elicitors has greater benefit for increasing cambium activity, making heartwood prematurely, and inducing the synthesis of sandal oil. However, heartwood formation is a long and complex physiological process that is regulated by spatial and temporal cues and involves multiple signaling crosstalking. Further work is needed to trace the exact function of BA and H_2_O_2_ on sesquiterpene synthesis during transcriptional and post-transcriptional regulation.

## Figures and Tables

**Figure 1 plants-10-00339-f001:**
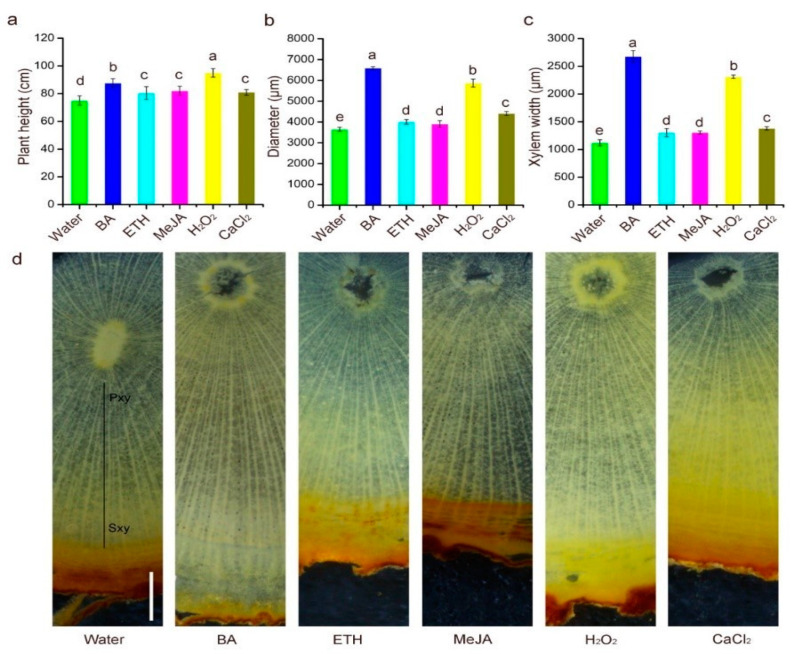
Exogenously applied chemical elicitors enhanced xylem development in stems of 1-year-old sandal trees. Plant height (**a**), mature stem diameter (**b**), and quantification of xylem width (**c**) in stems treated with chemical elicitors or water (control). Cross-sections of the 20th internode of stems treated with water, BA, ETH, MeJA, H_2_O_2_, and CaCl_2_ (**d**). Intact stem regions are shown and xylem width (black) are marked. Bars = 500 μm. Error bars = standard deviations. Different letters above graph bars indicate significant differences between means according to Duncan’s multiple range test (*p* < 0.05, *n* = 5). Pxy, primary xylem; Sxy, secondary xylem.

**Figure 2 plants-10-00339-f002:**
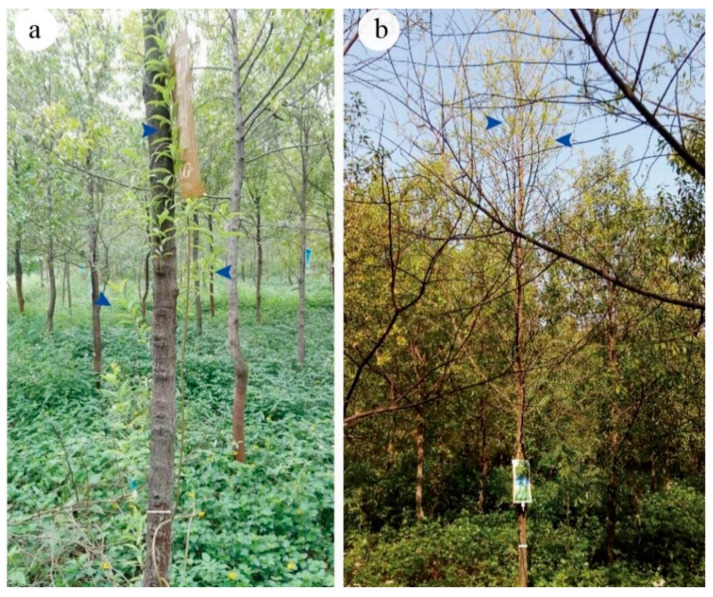
Morphology of leaves and stems of 5-year-old sandal trees treated with BA (**a**) and ETH (**b**) after one year. Blue arrows indicated in newly axillary branches and in b, senesced leaves and new developing axillary branches. White bars = 7 cm in (**a**,**b**) indicate stem width.

**Figure 3 plants-10-00339-f003:**
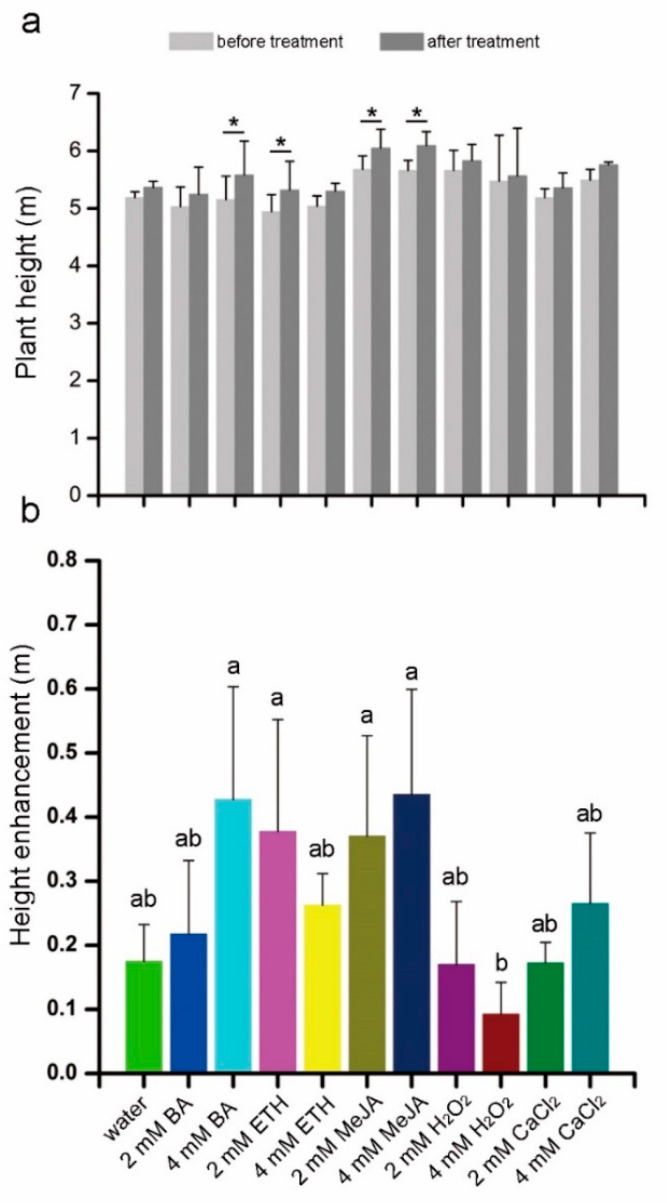
Plant height of 5-year-old sandal trees before and after injection of exogenously applied chemical elicitors after one year. (**a**), Plant height and (**b**) increased height in response to water (control) and several chemical elicitors. Error bars = standard deviation. In (**a**), * indicates significant differences between means of the before and after treatments according to a student’s *t*-test (*p* < 0.05, *n* = 4). Different letters in (**b**) indicate significant differences between means according to Duncan’s multiple range test (*p* < 0.05, *n* = 4).

**Figure 4 plants-10-00339-f004:**
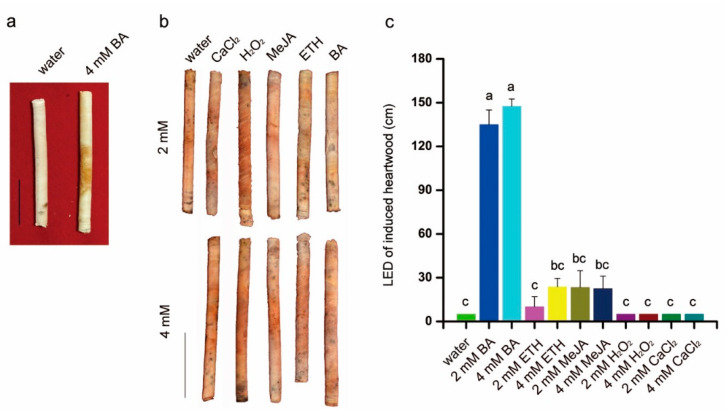
Exogenously applied chemical elicitors promoted heartwood formation of 5-year-old sandal trees. (**a**), Heartwood was obtained at a distance of 30 cm above the point of injection after treatment with 4 mM BA after 2 months. Pale-yellow heartwood formed in the middle of the stem in BA-treated trees while heartwood treated with water showed no obvious change in color. Bar = 2 cm. (**b**), Heartwood obtained at a distance of 30 cm above the injection point in trees treated with different chemical elicitors (CaCl_2_, H_2_O_2_, MeJA, ETH, and BA), or water, after 1 year. Bar = 2 cm; (**c**), Longitudinal heartwood extension (LED) of heartwood in response to exogenously applied chemical elicitors. Bar = 2 cm. Error bars = standard deviation. Different letters indicate significant differences between means according to Duncan’s multiple range test (*p* < 0.05, *n* = 4).

**Figure 5 plants-10-00339-f005:**
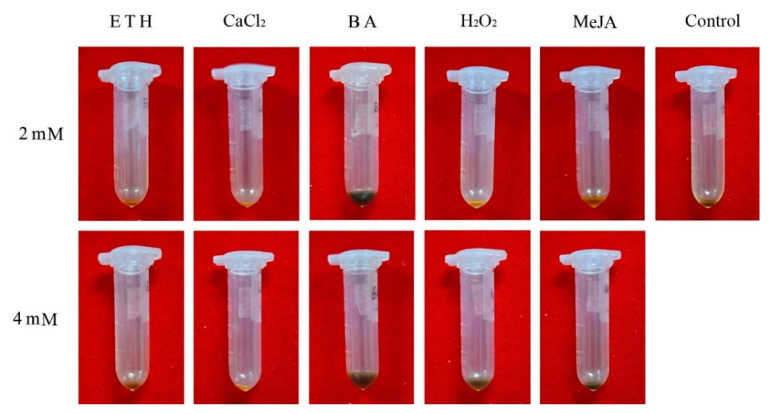
The effect of different chemical elicitors at 2 or 4 mM on concrete oil content of heartwood from 5-year-old sandal trees.

**Figure 6 plants-10-00339-f006:**
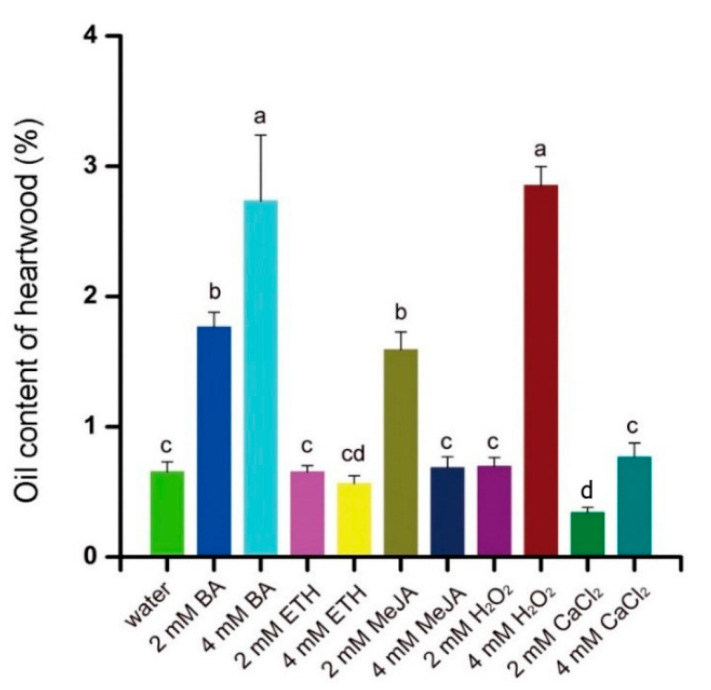
The effect of different chemical elicitors at 2 mM or 4 mM on concrete oil content of heartwood from 5-year-old sandal trees. Different letters indicate significant differences between means according to Duncan’s multiple range test (*p* < 0.05, *n* = 4).

**Figure 7 plants-10-00339-f007:**
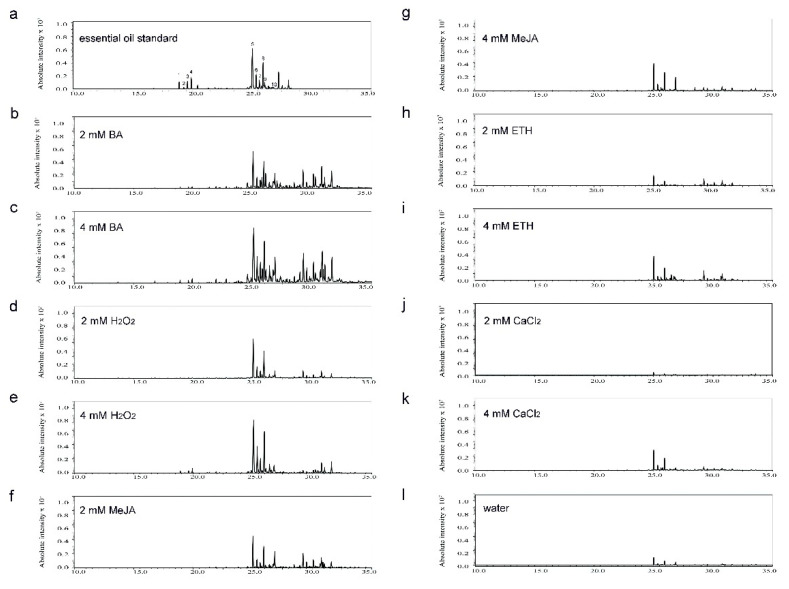
A total ions chromatogram of sandal concrete oil and mass spectrometric data from 5-year-old sandalwood trees in response to different treatments with essential oil standard, water (control) and different chemical elicitors at 2 or 4 mM. (**a**) Essential oil standard; (**b**) 2 mM BA; (**c**) 4 mM BA; (**d**) 2 mM H_2_O_2_; (**e**) 4 mM H_2_O_2_; (**f**) 2 mM MeJA; (**g**) 4 mM MeJA; (**h**) 2 mM ETH; (**i**) 4 mM ETH; (**j**) 2 mM CaCl_2_; (**k**) 4 mM CaCl_2_; (**l**) Water control.

**Figure 8 plants-10-00339-f008:**
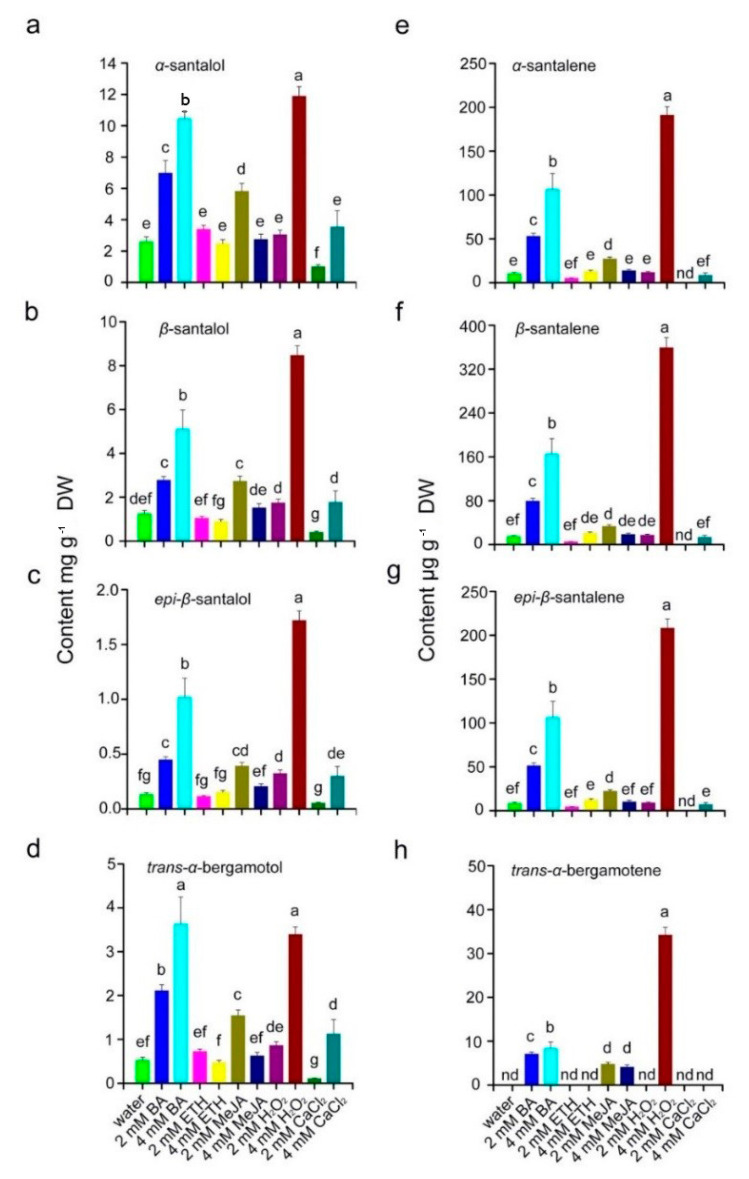
Production of major sesquiterpenoids in the heartwood from 5-year-old sandal trees in response to treatments with different chemical elicitors at 2 or 4 mM. *α*-santalol (**a**), *β*-santalol (**b**), *epi*-*β*-santalol (**c**), *trans*-*α*-bergamotol (**d**), *α*-santalene (**e**), *β*-santalene (**f**), *epi*-*β*-santalene (**g**) and *trans*-*α*-bergamotene (**h**). In (**a**) to (**d**), the unit is mg^−1^ dry weight (DW). In (**e**) to (**h**), the unit is μg^−1^ DW. Different letters indicate significant differences between means according to Duncan’s multiple range test (*p* < 0.05, *n* = 4). nd, not detected.

**Table 1 plants-10-00339-t001:** Effects of exogenously applied chemical elicitors on sandalwood stem diameter at a height of 30 and 130 cm above the ground.

Treatment	Stem Diameter at a Height of 30 cm above the Ground (cm)			Stem Diameter at a Height of 130 cm above the Ground (cm)		
	Before Treatment (Mean ± SD)	After Treatment (Mean ± SD)	Change Range * (Mean ± SD)	Before Treatment (Mean ± SD)	After Treatment (Mean ± SD)	Change Range * (Mean ± SD)
2 mM BA	7.51 ± 0.33	8.18 ± 0.31	0.67 ± 0.14 a	6.10 ± 0.16	6.76 ± 0.27	0.66 ± 0.17 ab
4 mM BA	7.61 ± 0.42	8.45 ± 0.45	0.85 ± 0.05 a	6.60 ± 0.27	7.33 ± 0.26	0.74 ± 0.15 ab
2 mM ETH	7.30 ± 0.11	7.73 ± 0.11	0.45 ± 0.10 ab	6.01 ± 0.08	6.68 ± 0.11	0.67 ± 0.09 ab
4 mM ETH	6.75 ± 0.25	6.69 ± 0.30	0.23 ± 0.05 b	5.99 ± 0.23	6.28 ± 0.24	0.30 ± 0.09 b
2 mM MeJA	7.29 ± 0.32	8.06 ± 0.58	0.78 ± 0.28 a	6.26 ± 0.31	7.06 ± 0.29	0.80 ± 0.06 ab
4 mM MeJA	7.81 ± 0.02	8.70 ± 0.09	0.90 ± 0.09 a	6.75 ± 0.04	7.39 ± 0.16	0.64 ± 0.14 ab
2 mM H_2_O_2_	7.57 ± 0.19	8.47 ± 0.23	0.90 ± 0.11 a	6.93 ± 0.51	7.42 ± 0.83	0.49 ± 0.17 ab
4 mM H_2_O_2_	7.55 ± 0.28	8.41 ± 0.14	0.85 ± 0.23 a	6.58 ± 0.45	7.10 ± 0.65	0.52 ± 0.23 ab
2 mM CaCl_2_	7.88 ± 0.55	8.55 ± 0.60	0.70 ± 0.09 a	6.68 ± 0.28	7.45 ± 0.27	0.77 ± 0.20 ab
4 mM CaCl_2_	7.87 ± 0.33	8.75 ± 0.44	0.85 ± 0.12 a	6.67 ± 0.18	7.76 ± 0.42	1.10 ± 0.25 a
Water control	7.69 ± 0.22	8.52 ± 0.28	0.83 ± 0.10 a	6.55 ± 0.11	7.25 ± 0.08	0.70 ± 0.19 ab

* Different letters indicate significant differences between means according to Duncan’s multiple range test (*p* < 0.05, *n* = 4).

## Data Availability

All data generated or analyzed during this study are included in this article.

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
