# Peer review of "Elicitors Modulate Young Sandalwood (Santalum album L.) Growth, Heartwood Formation, and Concrete Oil Synthesis"

_plants, 2021, doi:10.3390/plants10020339_

Round 1

Author Response

We have ansower all the questions of the reviewer 1

Reviewer 2 Report

The manuscript is correctly written, I have not observed errors or badly constructed sentences.
The abstract is well written and shows the work done. The introduction is very complete and shows that the authors have reviewed the published work. The references are correctly given and well used. The methods are well written and explained; The results have been analyzed in depth and show well the interpretation made of the data obtained. Perhaps the production variations of the essential oil components should be better completed.

Author Response

We have ansower all the questions of the reviewer 2, Please see the attched file.

Reviewer 3 Report

The authors done a comprehensive study on the effect of five chemical
elicitors, including 6-benzyladenine, ethephon, methyl jasmonate, hydrogen peroxide (H2O2) and calcium chloride (CaCl2) on growth, heartwood formation, and essential oil synthesis of young (1- and 5- year-old) sandalwood trees by integrating anatomical, physiological and mass spectrum evidence. The manuscript is closely related to the journal's topics. I suggest some minor notes on the manuscript.

I suggest adding some result in the abstract.

At the end of introduction, I suggest writing chemical elicitors like this “6-benzyladenine (BA), ethephon (ETH) and methyl jasmonate (MeJA)” and after you can use only abbreviation.

Why authors doesn't use the classic method for extracting EOs, is this method special for Heartwood EO.

Is this oil having only sesquiterpenoid? If no, please give the % of other constituents.

Author Response

We have ansower all the questions of the reviewer 3. Please see the attached file.

Round 2

Reviewer 1 Report

The authors did not answer all questions raised by this reviewer. They failed to provide sound explanations or convincing evidence concerning important issues.

The analytical methodology and obtained data are very questionable.

Author Response

Thank you so much for your suggestions. Here basis on you suggestion, We have rewrriten the text again. We have supplemented two new Figures in the text and make the reader clearly understand what we have done. Please see the attached newly revision.

One figure is our extracted oil in the tubes;

another figure is the GC-MS results.

Round 3

Reviewer 1 Report

None

Author Response

I have revised the text according to your suggestions. Thank you so much.
